# Selective Targeting of Cancer-Related G-Quadruplex Structures by the Natural Compound Dicentrine

**DOI:** 10.3390/ijms24044070

**Published:** 2023-02-17

**Authors:** Chiara Platella, Francesca Ghirga, Domenica Musumeci, Deborah Quaglio, Pasquale Zizza, Sara Iachettini, Carmen D’Angelo, Annamaria Biroccio, Bruno Botta, Mattia Mori, Daniela Montesarchio

**Affiliations:** 1Department of Chemical Sciences, University of Naples Federico II, Complesso Universitario di Monte S. Angelo, Via Cintia, 21, 80126 Napoli, Italy; 2Department of Chemistry and Technology of Drugs, Sapienza University of Rome, Piazzale Aldo Moro 5, 00185 Rome, Italy; 3Translational Oncology Research Unit, IRCCS—Regina Elena National Cancer Institute, Via Elio Chianesi 53, 00144 Rome, Italy; 4Department of Biotechnology, Chemistry and Pharmacy, University of Siena, Via Aldo Moro 2, 53100 Siena, Italy

**Keywords:** Dicentrine, G-quadruplex, natural compounds, telomeres, oncogenes

## Abstract

Aiming to identify highly effective and selective G-quadruplex ligands as anticancer candidates, five natural compounds were investigated here, i.e., the alkaloids Canadine, D-Glaucine and Dicentrine, as well as the flavonoids Deguelin and Millettone, selected as analogs of compounds previously identified as promising G-quadruplex-targeting ligands. A preliminary screening with the G-quadruplex on the Controlled Pore Glass assay proved that, among the investigated compounds, Dicentrine is the most effective ligand of telomeric and oncogenic G-quadruplexes, also showing good G-quadruplex vs. duplex selectivity. In-depth studies in solution demonstrated the ability of Dicentrine to thermally stabilize telomeric and oncogenic G-quadruplexes without affecting the control duplex. Interestingly, it showed higher affinity for the investigated G-quadruplex structures over the control duplex (K_b_~10^6^ vs. 10^5^ M^−1^), with some preference for the telomeric over the oncogenic G-quadruplex model. Molecular dynamics simulations indicated that Dicentrine preferentially binds the G-quadruplex groove or the outer G-tetrad for the telomeric and oncogenic G-quadruplexes, respectively. Finally, biological assays proved that Dicentrine is highly effective in promoting potent and selective anticancer activity by inducing cell cycle arrest through apoptosis, preferentially targeting G-quadruplex structures localized at telomeres. Taken together, these data validate Dicentrine as a putative anticancer candidate drug selectively targeting cancer-related G-quadruplex structures.

## 1. Introduction

The noncanonical secondary structures of nucleic acids have attracted increasing attention as potential drug targets for their involvement in various crucial biological processes [1]. Even if DNA within cells mainly exists in the double-stranded B-form, single strands of DNA can fold into various conformations [1,2]. Among them, a growing interest is directed at G-quadruplex structures formed by single-stranded, guanine-rich sequences. G-quadruplexes consist of cyclic, planar arrays of four guanines, called G-tetrads, and sequences connecting the tracts of adjacent guanines, called loops [3]. These structures are involved in DNA damage and genome instability in addition to key cancer-related cellular pathways [3,4]. Indeed, G-quadruplexes are mainly found in telomeres and oncogene promoters, and several experiments have ascertained that their formation in DNA is modulated during the cell cycle [3,4]. Therefore, some innovative anticancer strategies rely on G-quadruplex structure stabilization induced by small molecules at telomeres and oncogene promoters in order to block telomerase activity and oncogene expression, respectively [3,5,6,7], thus developing targeted therapies potentially endowed with low-to-null side effects. To date, about 3000 different G-quadruplex ligands have been identified [8], some of which show potential anticancer activities, but only a few have progressed to clinical trials due to selectivity issues [5]. DNA in duplex form is present in chromosomes in large excess compared with G-quadruplex structures. Therefore, a stringent ability to selectively recognize G-quadruplexes discriminating duplex DNA is one of the main requirements in the design of G-quadruplex ligands as potential anticancer drugs. 

In this context, we recently selected 28 natural compounds as putative G-quadruplex ligands using the docking-based virtual screening of an in-house library of ca. 1000 natural compounds and then experimentally evaluated their interaction with G-quadruplex structures [9]. Finally, 5 out of the 28 natural compounds were identified as the best G-quadruplex ligands in vitro, i.e., Bulbocapnine, Chelidonine, Ibogaine, Rotenone and Vomicine. 

Aiming to discover more effective and selective G-quadruplex ligands, a focused set of structural analogs of the previously identified hit compounds was here investigated, including the natural alkaloids Canadine, D-Glaucine and Dicentrine and the flavonoids Deguelin and Millettone (Table 1). These compounds have well-recognized biological and pharmacological properties but have never been deeply investigated as putative G-quadruplex ligands or, more generally, as DNA ligands.

Here, a preliminary screening was carried out with the aid of the G-quadruplex on the Controlled Pore Glass (G4-CPG) assay [10,11] in order to assess the capacity of the examined natural compounds to interact with telomeric and oncogenic G-quadruplexes, as well as their G-quadruplex vs. duplex selectivity. The compound showing the highest affinity and selectivity for the G-quadruplexes, i.e., Dicentrine, was then studied in its interaction in solution with the DNA targets of choice using multiple biophysical techniques. Additionally, structural details of the interaction of Dicentrine with G-quadruplex structures were obtained using molecular dynamics (MD) simulations. Finally, Dicentrine was evaluated for its anticancer activity as well as its ability to target G-quadruplex structures in cells via biological assays.
ijms-24-04070-t001_Table 1Table 1Chemical structures and features of the natural alkaloids Canadine, D-Glaucine and Dicentrine and the flavonoids Deguelin and Millettone.CompoundCommon Name (Library Code)Chemical StructureM. W. Molecular Formula SourceReference**Alkaloids****1**Canadine (BBN142)
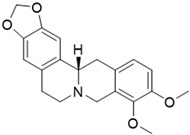
339.39C_20_H_21_NO_4_*Mahonia aquifolum* (Berberidaceae family)[12]**2**D-Glaucine (BBN45)
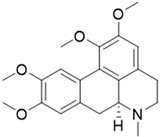
355.43C_21_H_25_NO_4_*Glaucium flavum* (Papaveracee family)[13]**3**Dicentrine (BBN172)
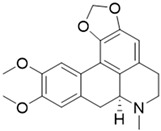
339.39C_20_H_21_NO_4_*Lindera megaphylla* (Lauraceae family)[14]**Flavonoids****4**Deguelin (BBN238)
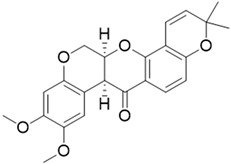
394.42C_23_H_22_O_6_*Mundulea sericea*(Fabaceae family)[15]**5**Millettone (BBN271)
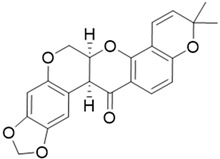
378.38C_22_H_18_O_6_*Derris trifoliata Lour*(Leguminosae family)[16]

## 2. Results and Discussion

### 2.1. Properties, Origin and Structural Features of the Selected Natural Compounds

The three alkaloids Canadine, D-Glaucine and Dicentrine are analogs of Bulbocapnine, an inhibitor of acetyl- and butyrylcholinesterase with dose-dependent activity [17]. Canadine is a benzylisoquinoline alkaloid that displays interesting antioxidant and antibacterial activities [18]. Correché et al. showed that Canadine has a strong antioxidant capacity against the oxidative stress induced by *tert*-butylhydroperoxide, a model inducer of endogenous reactive oxygen species (ROS) formation [12]. Canadine protects microsomal lipids from peroxidation, one of the reactions triggered by the formation of free radicals in cells and tissues. Furthermore, Chlebek et al. studied Canadine activity in human acetyl- and butyrylcholinesterase, involved in Alzheimer’s disease, showing an IC_50_ value in the low micromolar range [19].

D-Glaucine is a tetrahydroisoquinoline derivative long used as a remedy for coughs and other illnesses. D-Glaucine was identified as a relatively potent and selective inhibitor of soluble phosphodiesterase 4 (PDE4) isolated from bovine aortic muscle. PDE4 is the major isoenzyme present in human polymorphonuclear leukocytes, and its inhibition leads to increased cyclic AMP levels and the subsequent inhibition of several functional responses [20]. Moreover, D-Glaucine can inhibit the migration and invasion of human breast cancer cells [21].

Dicentrine is an aporphine-type isoquinoline alkaloid isolated from several medicinal plants, some of which are used in traditional medicine to treat cancer and other diseases [14]. Dicentrine has been shown to exert cytotoxic activity toward cancer cells by unwinding DNA and inhibiting the catalytic activity of DNA topoisomerases [22]. Konkimalla et al. have also shown that Dicentrine binds with high affinity to the erlotinib-binding site of EGFR—the epidermal growth factor receptor gene family, which evolved as important factors in cancer prognosis—by means of in silico modeling and the virtual screening approach [23]. Moreover, Dicentrine was found to stabilize a tetramolecular telomeric G-quadruplex model and to inhibit telomerase activity [24]. Furthermore, an in vitro tumor-growing assay in Severe Combined Immunodeficiency (SCID) mice showed that the intraperitoneal injection of Dicentrine at a 100 µg dose twice a week for 4 weeks significantly inhibited the tumor incidence of leukemia cell line K562 in SCID mice [14].

Deguelin and Millettone belong to the rotenoid family [16]. In particular, Deguelin is a rotenoid that shows high potential as a chemopreventive and therapeutic agent in vitro and in vivo against several types of cancer, including colon, lung and breast tumors. The anticancer effect of Deguelin is associated with cell cycle arrest and apoptosis via the downregulation of specific cell survival proteins, including Akt and mitogen-activated protein kinase. Deguelin is able to suppress HIF-1a expression and Hsp90 interaction in radioresistant lung cancer cells [15,25]. Finally, no in-depth studies on Millettone have been reported thus far.

### 2.2. Screening of Natural Compounds with the G4-CPG Assay

The G4-CPG assay is an affinity chromatography-based method conceived to improve the previously developed G-quadruplex on Oligo Affinity Support (G4-OAS) [11,26,27], allowing for the rapid and easy selection of G-quadruplex-specific ligands. It involves flowing solutions of putative G-quadruplex ligands through a Controlled Pore Glass (CPG) support functionalized with G-quadruplex-forming oligonucleotides. The compounds with high affinity for the G-quadruplex are retained, while those with low-to-null affinity are eluted. All the assay steps are monitored with UV measurements [10,11]. The two following cancer-related G-quadruplex-forming DNA sequences were chosen here as targets: (i) tel_26_, a 26-mer oligonucleotide of sequence d(TTAGGGTTAGGGTTAGGGTTAGGGTT), able to fold into a hybrid G-quadruplex, with its sequence extracted from the human 3′-telomeric overhang [28], and (ii) c-myc, a 33-mer oligonucleotide able to fold into a parallel G-quadruplex of sequence d(TGGGGAGGGTGGGGAGGGTGGGGAAGGTGGGGA), reproducing the regulatory region of the gene coding for the transcription factor C-MYC [29]. All the experiments carried out on the G-quadruplex targets were performed in parallel on a 27-mer unimolecular duplex-forming DNA sequence, hereafter named ds_27_, to verify if the analyzed compounds could discriminate G-quadruplex vs. duplex DNA. In particular, the duplex of choice consists of two self-complementary tracts, each containing the Dickerson sequence d(CGCGAATTCGCG), i.e., one of the best-studied models for B-DNA [30], connected by a TTT loop so as to obtain a hairpin-like structure. 

The protocol used for the binding assays has been previously reported [10] and is described here in the experimental section. Briefly, the compounds were initially evaluated for their solubility at the concentration and in the washing/releasing solutions used in the G4-CPG assay, proving to be fully soluble and stable in the experimental assay conditions. For each of the five natural compounds, tests were carried out to first evaluate unspecific binding on nude CPG support and then the ability to bind G-quadruplexes- and duplex-forming DNA oligonucleotide sequences. The results of the G4-CPG assay are summarized in Table 2. No significant unspecific binding was observed for all the tested compounds, thus allowing for their further analyses of the oligonucleotide-functionalized supports. Overall, D-Glaucine, Deguelin and Millettone showed a low-to-null ability to interact with G-quadruplex- and duplex-functionalized supports. On the other hand, Canadine showed good affinity and selectivity for G-quadruplex-functionalized supports; however, it was associated with significant binding to the nude CPG. Conversely, Dicentrine showed a low affinity for the duplex-functionalized support but high affinity toward the G-quadruplex-functionalized supports, with an interesting binding preference for the hybrid tel_26_ G-quadruplex.

In summary, within the investigated set of natural compounds, Dicentrine was the best G-quadruplex ligand in terms of target affinity and G-quadruplex vs. duplex selectivity according to the results of the G4-CPG assay, showing a two-fold higher affinity for the G-quadruplex targets than the control duplex. Therefore, it was advanced to further studies in solution to investigate its interaction with G-quadruplex structures using biophysical techniques.

### 2.3. Circular Dichroism Studies of Dicentrine with G-Quadruplexes and Control Duplex

The ability of Dicentrine to interact in solution with a telomeric and an oncogenic G-quadruplex model, as well as a control duplex, was investigated with circular dichroism (CD) experiments. The human telomeric G-quadruplex tel_26_ [28]; the oncogenic G-quadruplex Pu22T14T23 (hereafter also named Pu22), which is a shorter variant (22-mer) of the c-myc oligonucleotide of sequence d(TGAGGGTGGGTAGGGTGGGTAA) [31,32]; and the Dickerson dodecamer d(CGCGAATTCGCG) (ds_12_) [33] were used as valuable G-quadruplex and duplex models for studies in solution. The G-quadruplex-forming oligonucleotides, as well as the control duplex, were prepared by overnight-annealing the proper solutions at a 2 μM DNA concentration in 5 mM KCl, 5 mM KH_2_PO_4_, 5% DMSO buffer (pH 7). According to the literature, in these conditions, (i) tel_26_ folds into a hybrid 2-type G-quadruplex, featured by a double hump-band with maxima centered at 265 and 290 nm [34]; (ii) Pu22 exhibits a maximum centered at 265 nm and a minimum at 245 nm, typical of a parallel G-quadruplex [32]; and (iii) ds_12_ shows a positive band at 280 nm, along with an intense minimum at 251 nm, characteristic of a B-DNA duplex structure [35].

These oligonucleotides were titrated with increasing amounts of Dicentrine (up to 10 molar equivalents), and the corresponding CD spectra were recorded after each addition (Figure 1). In parallel, since Dicentrine has one chiral center and shows CD signals in the same range as the investigated oligonucleotide sequences, the spectra of Dicentrine at increasing amounts were recorded in the same buffer used for the titration experiments (Figure 1A), and the contribution of Dicentrine alone was subtracted from the spectra of the titrations of oligonucleotides with this alkaloid, as reported in Figure 1B–D. 

In detail, upon titration with Dicentrine, the tel_26_ G-quadruplex showed an increase in the CD signal intensity of the 290 nm band (Figure 1B), whereas, for the Pu22 G-quadruplex, a decrease in the intensity of the 265 nm band was observed (Figure 1C). Finally, the titration of ds_12_ with Dicentrine caused a slight increase in the intensity of the 251 nm band, accompanied by a dose-dependent increase in the 280 nm band (Figure 1D).

CD melting experiments were also performed on all the DNA/ligand mixtures to evaluate if the incubation with the ligand produced stabilizing effects on the G-quadruplex and duplex structures. The CD melting curves of tel_26_ G-quadruplex, Pu22 G-quadruplex and ds_12_ duplex in the absence or presence of Dicentrine (DNA/ligand 1:10 ratio) were recorded by following the changes in the wavelength of the characteristic CD intensity maximum/minimum of each DNA secondary structure (290, 265 and 251 nm for tel_26_, Pu22 and ds_12_, respectively) (Figure 2). Melting temperature (T_m_) values of 40, 79 and 63 °C were found for free tel_26_ G-quadruplex, Pu22 G-quadruplex and ds_12_ duplex, respectively (Table 3). Notably, stabilizing effects on both the tel_26_ and Pu22 G-quadruplexes were found (ΔT_m_ = +3 and +5 °C, respectively) (Figure 2A,B and Table 3), while no significant stabilizing effect on the ds_12_ duplex was observed in the presence of Dicentrine (ΔT_m_ = +1 °C) (Figure 2C and Table 3).

Overall, the CD titration experiments proved that, upon the binding of Dicentrine, the main folds of the investigated oligonucleotides were preserved, even if slight conformational changes were observed in all cases. Moreover, CD melting experiments demonstrated that Dicentrine positively affected the stability of G-quadruplex-forming oligonucleotides, with no relevant effect on the control duplex, thus further confirming the good G-quadruplex vs. duplex selectivity of this compound, as already observed with the G4-CPG assay.

### 2.4. Gel Electrophoresis Studies of Dicentrine with G-Quadruplexes and Control Duplex

To obtain further information on the binding of Dicentrine to the G-quadruplex targets, gel electrophoresis experiments were carried out on tel_26_, Pu22 and ds_12_. Native polyacrylamide gel electrophoresis (PAGE) experiments were performed, analyzing the oligonucleotide samples at a 2 µM concentration mixed with different Dicentrine amounts (from 1:0 to 1:10 DNA/ligand ratio) in 5 mM KCl, 5 mM KH_2_PO_4_, 5% DMSO buffer (Figure 3). Furthermore, a step ladder was used as a size marker with 5 bp differences between adjacent DNA fragments.

Native PAGE experiments showed a single band for tel_26_ G-quadruplex in the absence of Dicentrine: the comparison with the migration of the ladder fragments confirmed a monomeric G-quadruplex fold for this oligonucleotide (Figure 3A). The addition of 1 to 10 molar equivalents of Dicentrine with respect to the DNA did not detectably affect the mobility of the oligonucleotide, suggesting that the interaction with the ligand did not alter the tel_26_ G-quadruplex monomeric fold (Figure 3A).

As regards the Pu22 G-quadruplex, a single band was observed in the absence of Dicentrine, and even in this case, no variation in the oligonucleotide migration was detected at the different Pu22 G-quadruplex/Dicentrine ratios explored (Figure 3B).

Finally, gel electrophoresis studies on free ds_12_ duplex revealed the presence of two bands: one main band with lower mobility, associated to the bimolecular duplex fold, and one less intense band with higher mobility, attributable to the single DNA strands derived from the partial unfolding of the duplex (Figure 3C). A slight increase in the single-strand band intensity was observed upon the addition of Dicentrine, suggesting a partial unwinding of the bimolecular duplex induced by the ligand, in agreement with a previous report [22].

Overall, in full agreement with the CD titration results, the PAGE experiments proved that the main conformations of the investigated DNA secondary structures were preserved upon Dicentrine binding.

### 2.5. Fluorescence Spectroscopy Studies of Dicentrine with G-Quadruplexes and Control Duplex

The fluorescence spectrum of a 2 μM solution of Dicentrine was recorded in 5 mM KCl, 5 mM KH_2_PO_4_, 5% DMSO buffer (pH 7). Dicentrine showed one strong emission band at 367 nm (Figure 4, black lines). Thus, the interaction of the ligand with tel_26_ G-quadruplex, Pu22 G-quadruplex and ds_12_ duplex was studied using fluorescence spectroscopy through reverse titration experiments. In detail, fluorescence titrations were carried out at a fixed concentration of the ligand (i.e., 2 μM) by adding increasing amounts of tel_26_, Pu22 or ds_12_, previously annealed in 5 mM KCl, 5 mM KH_2_PO_4_, 5% DMSO buffer (pH 7). Upon each addition, the corresponding fluorescence spectrum was recorded after stabilizing the signal. The fraction of bound ligand was calculated from the obtained fluorescence intensity values and plotted as a function of the DNA concentration. These data were then fitted with the independent and equivalent-sites model [36] in order to derive the related binding constants and stoichiometries.

Significant fluorescence quenching was observed upon the titration of Dicentrine with all the investigated oligonucleotides (Figure 4, left panels). Fitting curves related to the fluorescence data for Dicentrine with the tel_26_ and Pu22 G-quadruplexes and the ds_12_ duplex are shown in Figure 4, right panels. Notably, some preference for the hybrid tel_26_ G-quadruplex (K_b_ = (2.8 ± 1.2) × 10^6^ M^−1^) compared with the parallel Pu22 G-quadruplex (K_b_ = (0.8 ± 0.5) × 10^6^ M^−1^) was observed, in line with the affinity trend obtained with the G4-CPG assay. In addition, a good G-quadruplex vs. duplex selectivity was found, considering the significant difference of one order of magnitude in their binding constants (K_b_ = (3.2 ± 0.4) × 10^5^ M^−1^), in full agreement with the G4-CPG assay. Binding stoichiometries of 1:1, 1:2 and 1:5 were obtained for the tel_26_ G-quadruplex/Dicentrine, Pu22 G-quadruplex/Dicentrine and ds_12_ duplex/Dicentrine systems, respectively, clearly showing that their binding to G-quadruplex structures is more specific compared with duplex DNA.

### 2.6. Molecular Modeling

The binding mode of Dicentrine to target G-quadruplexes tel_26_ and Pu22 was investigated using a combination of molecular docking and molecular dynamics (MD) simulations. According to our previous study on the parent natural compounds [9], Dicentrine was docked into the groove near the 3′-end of the first NMR model of the target G-quadruplex structures in the AutoDock program [37], and the docking complexes were relaxed for 500 ns in unrestrained MD simulations carried out on explicit solvent and K^+^ ions (Figure 5).

While molecular docking suggested that Dicentrine is potentially able to fit the groove of the target G-quadruplexes (binding energy to tel_26_ and Pu22 = −7.18 and −8.07 kcal/mol, respectively), differences in the binding mode to the tel_26_ and Pu22 G-quadruplexes were observed by MD simulations.

In the case of the tel_26_ G-quadruplex, MD simulations clearly showed that the compound binds stably and persistently to the binding site identified by docking with the groove of the G-quadruplex structure, in close proximity to G4-G6, T8 and G23-T25 (Figure 5A,C and Figure 6A–C). Within this binding site, Dicentrine interacts with the phosphate backbone of G5 from the middle G-tetrad and establishes a water-bridged H-bond with T8 from the loop (Figure 6B,C).

MD simulations of Pu22/Dicentrine interactions provided very interesting results. Indeed, while docking highlighted a putative binding site within the groove of the Pu22 G-quadruplex in proximity to G19 and T20, this site proved very unstable in MD simulations and was rapidly left by Dicentrine to move into the bulk solvent. However, the remarkable propensity of Dicentrine to bind the Pu22 G-quadruplex was highlighted by the fast transfer of the compound from the solvent to the G-quadruplex, where it preferentially recognizes the G-tetrad formed by G4, G8, G13 and G17 (Figure 6D). It is worth noting that this site was not accessible in docking simulations because it is occupied by A3 in the reference NMR structure used as a rigid receptor [39]. In MD simulations, Dicentrine displaces A3 and stacks on top of the G-tetrad in a sandwich-like conformation with G2 (Figure 6D,E). Additionally, Dicentrine interacts directly with the phosphate backbone of G2 and through water-bridged connections with G5 and G8 (Figure 6F). These results corroborate the relevance of MD simulations in the investigation of the binding mode of small molecules to G-quadruplex structures.

Based on the 1:2 DNA/ligand binding stoichiometry of Dicentrine to Pu22 G-quadruplex observed with fluorescence spectroscopy, intermolecular recognition was investigated using MD simulations by randomly placing two molecules of Dicentrine in the simulation box at a distance higher than 30 Å from the Pu22 G-quadruplex, as well as from each other. In agreement with the binding mode shown in Figure 6D–F, the results of these MD simulations clearly suggest that Dicentrine preferentially stacked onto the G-tetrad of the Pu22 G-quadruplex formed by G4, G8, G13 and G17, as evidenced by one of the two Dicentrine molecules in both MD replicas (Figure 7 and Figure 8). Notably, the primary binding site fully corresponds to the site observed in MD simulations carried out on the docking complex in a 1:1 stoichiometry (Figure 6D–F). On the other hand, the second Dicentrine molecule binds to different sites of the Pu22 G-quadruplex, including (i) the groove of the Pu22 G-quadruplex, including the loop T11-A12, by stacking on A12 (Figure 8A) and (ii) stacking on top of the first Dicentrine molecule on the G-tetrad at the 5′-end of the G-quadruplex in a sandwich-like conformation with G2 (Figure 8B). Interestingly, the interaction with these secondary sites seems to occur after the first Dicentrine molecule is bound, and it is apparently characterized by lower conformational stability (Figure 7B,C).

Finally, the relative binding energy of Dicentrine to the tel_26_ G-quadruplex and the primary and secondary sites of the Pu22 G-quadruplex identified with MD simulations was calculated using the Molecular Mechanics Generalized Born Surface Area (MM-GBSA) approach. The results clearly highlight that Dicentrine binding to the 5′-end G-tetrad of Pu22 is thermodynamically favored compared with binding to other sites, i.e., A12 or already-bound Dicentrine. The calculated delta energy of binding of Dicentrine to different systems is reported in Table 4 and corroborates the hypothesis that, considering the 1:2 DNA/ligand stoichiometry, the first molecule of Dicentrine binds with higher affinity to a primary site located on top of the G-tetrad at the 5′-end, while the second Dicentrine molecule binds to secondary sites with lower affinity and conformational stability. Notably, the delta energy of the binding of Dicentrine to the tel_26_ G-quadruplex is fully comparable to the one predicted for the interaction within the primary site of the Pu22 G-quadruplex, in agreement with the fluorescence data (Table 4).

Overall, these results provide a consistent structural explanation of the interaction of Dicentrine with the tel_26_ and Pu22 G-quadruplexes in solution, as observed with spectroscopic and electrophoretic techniques.

### 2.7. Biological Assays

Aiming to further validate Dicentrine as a putative selective antitumor candidate targeting G-quadruplex structures, in-cell assays were performed. Compared with the above experimental and in silico analyses, the use of biological models introduces an additional level of complexity, thus representing an important step in selecting candidate leads that are potentially suitable for preclinical and clinical trials.

In this regard, Dicentrine was first evaluated for its properties as an antitumor agent. In detail, human-transformed BJ fibroblasts (BJ-EHLT cells) were treated with increasing doses (from 1 to 50 µM) of Dicentrine and, after 72 h of treatment, cell viability was evaluated. Interestingly, the results of the crystal violet assay (Figure 9A, blue line) evidenced the dose-dependent activity of Dicentrine, with an IC_50_ value of about 25 µM.

Since anticancer drugs are much more effective if their selectivity for cancer versus normal cells is higher, viability experiments were also carried out in the normal counterpart of BJ-EHLT cells, i.e., immortalized BJ fibroblasts, BJ-hTERT. As evidenced by the growth curve, normal cells were almost insensitive to Dicentrine at all the evaluated doses, thus underlying the high selectivity of this G-quadruplex ligand against cancer cells (Figure 9A, orange line).

BJ-EHLT and BJ-hTERT, derived from the same cell line of origin (BJ fibroblasts), represent extremely valid models to test the selectivity of treatments against tumor cells. Nevertheless, the use of BJ-EHLT is frequently debated because these cells—obtained through sequential immortalization and transformation events—are considered an “artificial” tumor model. Based on these considerations, the effectiveness of Dicentrine treatment was also evaluated in HeLa, a human cervical cancer cell line. Notably, the crystal violet assay (Figure 9B), fully reproducing the results obtained in BJ-EHLT, confirmed the antitumor activity of Dicentrine (IC_50_ value of about 25 µM). 

In parallel, the effect of Dicentrine on cancer cells was also evaluated using a fluorescence-associated cell sorter (FACS). Interestingly, time-course cell cycle analyses (from 24 to 72 h) showed that the treatment of HeLa cells with different doses of Dicentrine (10, 25 and 50 µM) promoted time- and dose-dependent cell cycle arrest, evidenced by the accumulation of cells in the G_2_/M phase (Figure 10A). In addition, FACS analyses showed the appearance of a subG1 peak, indicative of the capability of Dicentrine to induce cell apoptosis in treated cells (Figure 10A). Moreover, Annexin V analyses confirmed the capability of Dicentrine to induce apoptosis in treated HeLa cells, reaching about 20% and 50% of Annexin V-positive cells upon treatment with 25 and 50 µM of Dicentrine for 72 h, respectively (Figure 10B).

Altogether, these results highlighted the effectiveness of Dicentrine in promoting a potent and selective antitumor activity by inducing cell cycle arrest through apoptosis.

In order to investigate the mechanisms by which Dicentrine promotes apoptosis in cancer cells, additional in vitro tests were carried out. In more detail, the capability of Dicentrine to stabilize G-quadruplex structures in cells was evaluated. For these experiments, HeLa cells were treated with different doses of Dicentrine (10, 25 and 50 µM) and, after 24 h of treatment, the G-quadruplex structures—detected with a specific antibody—were evaluated using immunofluorescence (IF) microscopy. In agreement with the biophysical results, Dicentrine was found to promote, in a dose-dependent manner, the stabilization of G-quadruplex structures in tumor cells (Figure 11).

Additionally, considering that the antitumor activity of G-quadruplex ligands is mainly due to their capability to induce DNA damage, the amount of phosphorylated histone H2AX (γH2AX), a typical hallmark of DNA double-strand breaks [40], was evaluated in the cells treated with Dicentrine. Interestingly, complementary technical approaches, based on both confocal microscopy (IF) and biochemical (Western blot) analyses, evidenced a robust increase in γH2AX levels in Dicentrine-treated cells compared with their untreated counterparts (Figure 12A,B), highlighting the efficacy of the treatment in inducing DNA damage in tumor cells. For completeness, we then evaluated whether Dicentrine-induced DNA damage was localized at the telomeres. Thus, a fluorescence in situ hybridization (FISH) assay was performed, aimed at defining the number of γH2AX spots colocalizing with a fluorescent telomeric probe (Telomere Induced Foci, TIF). Interestingly, quantitative confocal analyses evidenced that Dicentrine promotes a dose-dependent increase in the percentage of TIF-positive cells (TIF ≥ 4), thus demonstrating the effective activity of this G-quadruplex ligand at the telomeres (Figure 12C,D).

Based on this robust set of collected biological data, it is possible to conclude that Dicentrine—through its capability to preferentially target G-quadruplex structures localized at the telomeres—might represent a valid lead compound in developing a new class of potent and selective anticancer drugs.

## 3. Materials and Methods

### 3.1. Chemistry

All the tested compounds are known structures belonging to the inhouse library of natural products available from the Organic Chemistry Laboratory of the Department of Chemistry and Technology of Drugs of Sapienza University of Rome, Italy. The chemical identity of noncommercial compounds was assessed by rerunning NMR experiments and proved to be in agreement with the literature data [41].

Compound **1** (Canadine, or (S)-9,10-dimethoxy-5,8,13,13a-tetrahydro-6H-[1,3]dioxolo [4,5-g]isoquinolino [3,2-a]isoquinoline) was purchased from Chemische Fabrik Karl Bucher GmbH and used without further purification.

Compound **2** (D-Glaucine, or (S)-1,2,9,10-tetramethoxy-6-methyl-5,6,6a,7-tetrahydro-4H-dibenzo[de,g]quinoline) showed NMR spectra identical to those reported in the literature [41].

Compound **3** (Dicentrine, or (S)-10,11-dimethoxy-7-methyl-6,7,7a,8-tetrahydro-5H-[1,3]dioxolo [4’,5’:4,5]benzo [1,2,3-de]benzo[g]quinoline) was purchased from Biosynth Carbosynth and used without further purification.

Compound **4** (Deguelin, or (7aS,13aS)-9,10-dimethoxy-3,3-dimethyl-13,13a-dihydro-3H-pyrano [2,3-c:6,5-f’]dichromen-7(7aH)-one) was purchased from Biosynth Carbosynth and used without further purification.

Compound **5** (Millettone, or (1S,14S)-7,7-dimethyl-2,8,18,20,24-pentaoxahexacyclo [12.11.0.03,12.04,9.015,23.017,21]pentacosa-3(12),4(9),5,10,15,17(21),22-heptaen-13-one) was purchased from BioTeck and used without further purification.

### 3.2. G4-CPG Assay

Stock solutions (4 mM) for each compound were prepared by dissolving a known amount of the sample in pure DMSO. A measured volume was taken from the stock solution to obtain a 60 μM compound solution in 50 mM KCl, 10% DMSO, 10% CH_3_CH_2_OH aq. solution. The detailed general procedure adopted for the assays is described as follows: weighed amounts of the nude CPG and G-quadruplex-/duplex-functionalized CPG supports (ca. 8 mg) were left in contact with 300 μL of the compound solution in a polypropylene column (4 mL volume, Alltech) equipped with a polytetrafluoroethylene frit (10 μm porosity), a stopcock and a cap [10,11]. After incubation on a vibrating shaker for 4 min, each support was washed with defined volumes of the washing solution (50 mM KCl, 10% DMSO, 10% CH_3_CH_2_OH aq. solution) or the releasing solution (2.5 M CaCl_2_, 15% DMSO aq. solution or pure DMSO), and all the eluted fractions were separately analyzed with UV measurements. After treatment with the releasing solution, able to induce G-quadruplexes and hairpin duplex denaturation, the supports were suspended in the washing solution and then subjected to annealing by taking them at 75 °C for 5 min and then slowly cooling them to room temperature.

The UV measurements were performed on a JASCO V-550 UV-vis spectrophotometer. A quartz cuvette with a path length of 1 cm was used. The UV quantification of the compounds was determined by measuring the absorbance relative to the λ_max_ characteristic of each compound and referring it to the corresponding calibration curves. The errors associated with the % of bound ligands are within ±2%.

### 3.3. Circular Dichroism

CD spectra were recorded in a quartz cuvette with a path length of 1 cm on a Jasco J-715 spectropolarimeter equipped with a Peltier-type temperature control system (model PTC-348WI). The spectra were recorded at 20 °C in a range of 240–600 nm, with a 2 s response, a 200 nm/min scanning speed and a 2.0 nm bandwidth, and they were corrected by subtracting the background scan with buffer. All the spectra were averaged over 3 scans. The oligonucleotides d[(TTAGGG)_4_TT] (tel_26_), d(TGAGGGTGGGTAGGGTGGGTAA) (Pu22) and d(CGCGAATTCGCG) (ds_12_) were purchased from Biomers (www.biomers.net) as HPLC-purified compounds with a purity of > 99%. The oligonucleotides were dissolved in a 5 mM KCl, 5 mM KH_2_PO_4_, 5% DMSO buffer (pH 7) to obtain 2 μM solutions and then annealed by heating them at 95 °C for 5 min, followed by slow cooling to room temperature. CD titrations were obtained by adding increasing amounts of Dicentrine (up to 10 molar equivalents, corresponding to a 20 μM solution in ligand) to the buffer alone or to the oligonucleotide solutions. For the CD melting experiments, the ellipticity was recorded at 290 nm for tel_26_, 265 nm for Pu22 and 253 nm for ds_12_ with a temperature scan rate of 0.5 °C/min in a range of 10–95 °C.

### 3.4. Gel Electrophoresis

The oligonucleotide/Dicentrine mixtures (up to 1:10 ratio) and free oligonucleotide samples, together with an O’RangeRuler™ 5 bp DNA Ladder (ThermoFisher Scientific), were loaded and analyzed on native 20% polyacrylamide (19:1 acrylamide:bisacrylamide) gels. TBE 1× supplemented with 10 mM KCl was used as a running buffer. The oligonucleotide samples loaded on the gels were 2 µM in oligonucleotide concentration per strand in 5 mM KCl, 5 mM KH_2_PO_4_, 5% DMSO buffer. No migration marker was used. Samples were electrophoresed for 3 h at 100 V at room temperature. The bands were visualized using GelGreen™ staining.

### 3.5. Fluorescence Spectroscopy

Fluorescence spectra were recorded at 20 °C on a HORIBA JobinYvon Inc. FluoroMax^®^-4 spectrofluorometer equipped with an F-3004 Sample Heater/Cooler Peltier Thermocouple Drive by using a quartz cuvette with a 1 cm path length. For the fluorescence titration experiments with Dicentrine, an excitation wavelength of 307 nm was used, and the spectra were registered in a range of 320–600 nm.

Titrations were carried out at a fixed concentration (2.0 μM) of ligand. Increasing amounts of tel_26_, Pu22 or ds_12_ (up to 10 μM concentration) were added from 120 μM annealed stock solutions of each DNA sample dissolved in a 5 mM KCl, 5 mM KH_2_PO_4_, 5% DMSO buffer (pH 7). After each addition, the system was allowed to equilibrate for 10 min before recording the spectra.

The fraction of bound Dicentrine was calculated from a fluorescence intensity of 367 nm and reported in a graph as a function of the DNA concentration. The fraction of the bound ligand was determined using the equation
α = (Y − Y_0_)/(Y_b_ − Y_0_)
where Y, Y_0_ and Y_b_ are the values of fluorescence emission intensity at the maximum of each titrant concentration at the initial and final state of the titration. These points were fitted with an independent and equivalent-sites model [36] using the Origin 8.0 program.

The equation of the independent and equivalent-sites model is as follows:$$\alpha =\left(\frac{1}{2{\left[L\right]}_{0}}\right)\left\{\left({\left[L\right]}_{0}+n\left[\mathrm{DNA}\right]+\frac{1}{K_{b}}\right)-\sqrt{{\left({\left[L\right]}_{0}+n\left[\mathrm{DNA}\right]+\frac{1}{K_{b}}\right)}^{2}-4{\left[L\right]}_{0}n\left[\mathrm{DNA}\right]}\right\}$$
where α is the mole fraction of the ligand in the bound form; [L]_0_ is the total ligand concentration; [DNA] is the added DNA concentration; n is the number of equivalent and independent sites in the DNA structure; and K_b_ is the binding constant.

### 3.6. Molecular Modeling

Molecular docking and MD simulations were carried out according to the protocol previously described [9]. Briefly, molecular docking was carried out with AutoDock4 [37] using the first NMR model of target G-quadruplexes tel_26_ (PDB-ID: 2JZP) [38] and c-myc (Pu22 sequence) (PDB-ID:1XAV) [39] as rigid receptors. The binding site on tel_26_ was centered on the groove near the 3′-end in the region surrounded by A3-G6 and G23-T26, while the binding site on Pu22 was centered on the groove near the 3′-end in the region defined by the A3-G6 and G19-A22 nucleotides. MD simulations were run with Amber18, starting from the docking complexes [42], using the General Amber Force Field for the parametrization of Dicentrine and the OL15 force field for the parametrization of the G-quadruplexes [43,44,45]. Macromolecular complexes were solvated in a rectangular box of TIP3P-type water molecules, while K^+^ ions were added to neutralize the total charge of the system. Production of MD trajectories was extended up to 500 ns at constant pressure and without positional restraints for every MD simulation. For MD-based intermolecular recognition studies, the initial system was built by placing the molecule(s) of Dicentrine at a distance higher than 30 Å to each other and the G-quadruplex. Two independent MD replicas of 500 ns each were run to achieve statistical significance. Analysis of MD trajectories was carried out with CPPTRAJ, while the delta energy of the binding of Dicentrine to both tel_26_ and Pu22 was calculated with the MM-PBSA.py software [46,47].

### 3.7. Biological Experiments

#### 3.7.1. Cells and Culture Condition

Human fibroblasts (BJ) and human cervical cancer cells (HeLa) were purchased from American Type Culture Collection (ATCC, CRL-2522, CCL-2). BJ-hTERT cells were obtained by infecting primary BJ cells with a retrovirus carrying hTERT (Addgene plasmid #1773). BJ-EHLT cells were derived from the transformation of BJ fibroblasts with hTERT and SV40 early region [48]. BJ-hTERT, BJ-EHLT and HeLa were grown in Dulbecco Modified Eagle Medium (DMEM; EuroClone, Milan, Italy; ECM0728L) supplemented with 10% Fetal Bovine Serum (FBS, ThermoFisher Scientific/Gibco, Waltham, MA, USA; 10,270–106), 2 mM l-glutamine and antibiotics at 37 °C in a 5% CO_2–_95% air atmosphere.

#### 3.7.2. Viability Assay (Crystal Violet)

BJ-EHLT, BJ-hTERT and HeLa cells were seeded in 24-well plates at a density of 6 × 10^3^, 12 × 10^3^ and 8 × 10^3^ for each well, respectively. After 24 h, the cells were treated with different concentrations of Dicentrine (0, 1, 5, 10, 25, 50 µM) for 72 h. Upon treatment completion, the cell medium was removed, and the wells were washed twice in phosphate-buffered saline (PBS) and fixed with 4% formaldehyde for 15 min at room temperature (RT). After washing, 500 µL of crystal violet staining solution (Sigma-Aldrich, St. Louis, MO, USA) was added to each well and incubated for 30 min at RT. Finally, the plates were rinsed twice with water and air-dried, and cell pellets were dissolved in 400 µL of a 10% aqueous solution of acetic acid. In total, 200 µL of each sample was transferred to a 96-well plate, and the optical density was measured at 570 nm (OD_570_) with an ELISA reader (Thermo Scientific, Waltham, MA, USA). The average absorbance in each condition was used to calculate the viability expressed as a percentage of treated vs. untreated conditions.

#### 3.7.3. Flow Cytometry

For the analysis of the cells’ progression through the cell cycle phases, HeLa cells were treated with 10, 25 and 50 µM of Dicentrine for the indicated times. Both floating and adherent cells were harvested, washed twice with PBS 1× and finally fixed in 70% CH_3_CH_2_OH at 4 °C overnight. Next, cells were washed with PBS 1×, stained with 50 mg/mL of propidium iodide (PI; Invitrogen-eBioscience) in the presence of 75 kU/mL of RNase and analyzed using FACSCelesta (BD Biosciences, San Jose, CA, USA).

The evaluation of apoptosis was carried out following the Annexin V assay versus PI staining, as previously described [49]. Briefly, both floating and adherent cells were harvested and suspended in Annexin-binding buffer (final concentration of 1 × 10^6^ cells/mL). Thus, cells were incubated with fluorescein isothiocyanate–Annexin V (Annexin-FITC) and PI for 15 min at RT away from the light and immediately analyzed by using FACSCelesta (BD Biosciences, San Jose, CA, USA). For each condition, 20,000 events were measured, setting 488 and 525 nm as the excitation and emission wavelengths, respectively. Data obtained were analyzed with the FACS Diva software (BD Biosciences).

#### 3.7.4. Immunofluorescence (IF) Experiments

HeLa cells were seeded on glass coverslips in a 24-well plate at a density of 2 × 10^4^ cells/well. After 24 h of treatment with Dicentrine (0, 10, 25, 50 µM), cells were fixed in 4% formaldehyde in PBS (10 min at RT) and permeabilized with 0.25% Triton X-100 (5 min at RT). For immuno-labeling, cells were incubated with a mouse mAb that specifically recognizes DNA/RNA G-quadruplex structures (BG4 antibody; Absolute Antibody, Oxford, UK, #Ab00174-1.1). After 2 h of incubation at RT, cells were washed three times with PBS 1X and then incubated for an additional 1 h with an antibody anti-mouse IgG (H + L), F(ab’)2 Fragment (Alexa Fluor 555 Conjugate; Cell Signaling Technology, Danvers, MA, USA, #4409S). Nuclei were stained with 4′,6-diamidino-2-phenylindole (DAPI; Sigma-Aldrich, D9542), and fluorescence signals were recorded by using a Zeiss Laser Scanning Microscope 510 Meta (63× magnification) (Zeiss, Jena, Germany).

IF experiments were quantified using ImageJ (version 1.53e, National Institutes of Health, NIH, Bethesda, MD, USA). At least 25 cells were screened for each condition, and the results were expressed as the fold change of the fluorescence intensity over the negative control. Histograms show the mean ± SD of three independent experiments.

#### 3.7.5. Fluorescence In Situ Hybridization (FISH) Assays

In order to evaluate the DNA damage, immunofluorescence experiments were combined with telomeric FISH assays. After fixation and permeabilization (as reported for the IF experiments), cells were blocked with 3% BSA in PBS 1× for 1 h and incubated with the mouse mAb anti-phospho-histone H2AX (γH2AX Ser139; Millipore, 05-636) at 4 °C overnight. After two washes with 0.05% Triton X-100 in PBS 1×, the cells were incubated with the secondary antibody anti-mouse IgG (H + L), F(ab’)2 Fragment (Alexa Fluor 488 Conjugate) for 1 h at RT and then washed twice with PBS 1×. Successively, cells were fixed in 4% formaldehyde (PBS 1X) again and subjected to standard telomere DNA FISH, as previously described [50]. Nuclei were stained with DAPI. For a quantitative analysis of γH2AX positivity, at least 250 cells/condition were scored in triplicate. For Telomere-Induced Foci (TIFs) analysis, at least 25 γH2AX-positive cells on a single plane were scored. Nuclei showing at least four colocalizations of a TelC-Cy3 telomeric probe (Panagene, Daejeon, Korea) and γH2AX were considered TIF-positive. Fluorescence signals were recorded with a Zeiss Laser Scanning Microscope 510 Meta (63× magnification) (Zeiss, Jena, Germany).

#### 3.7.6. Western Blot Analysis

Western blot analysis was performed as previously reported [51]. Briefly, HeLa cells were collected and resuspended in lysis buffer (50 mM Tris-HCl pH 7.5, 5 mM EDTA, 250 mM NaCl, 0.1% Triton) complemented with protease (Thermo Scientific, A32953) and phosphatase inhibitors (Thermo Scientific, 88667). Total proteins were fractionated using SDS-polyacrylamide gel electrophoresis and then transferred to a nitrocellulose membrane (Amersham, Arlington Heights, IL, USA). Membranes were probed with the following primary antibodies: mouse mAb anti-phospho-histone H2AX (γH2AX; Millipore, 05-636) and mouse mAb anti-β-actin (Sigma-Aldrich, A5441).

#### 3.7.7. Statistical Analyses

Experiments were replicated three times, and the data were expressed as means ± standard deviation (SD). GraphPad Prism 6 was used for the statistical analysis, and the differences between groups were analyzed using an unpaired Student’s *t*-test. Differences were considered statistically significant for * *p* < 0.05; ** *p*< 0.01; *** *p* < 0.001.

## 4. Conclusions

A stringent requirement for a G-quadruplex ligand to be putatively advanced to preclinical and clinical studies is the ability to selectively recognize G-quadruplexes by discriminating duplex DNA. In this context, we recently identified five natural compounds (i.e., Bulbocapnine, Chelidonine, Ibogaine, Rotenone and Vomicine) as efficient G-quadruplexes ligands with anticancer properties through a multidisciplinary approach combining computational, biophysical and biological studies. Here, aiming to improve the effectiveness and the G-quadruplex vs. duplex DNA selectivity of the previously identified hits, a set of their structural analogs was investigated, including the natural alkaloids Canadine, D-Glaucine and Dicentrine and the flavonoids Deguelin and Millettone. 

Within these natural compounds, Dicentrine emerged as the best ligand of telomeric and oncogenic G-quadruplexes in terms of affinity and G-quadruplex vs. duplex selectivity according to the results of the G4-CPG assay. Therefore, it was advanced to further studies in solution in its interaction with G-quadruplex structures to investigate its binding features in depth. 

CD and PAGE experiments proved that the main folds of the investigated oligonucleotides were preserved upon the binding of Dicentrine. In parallel, CD melting experiments demonstrated that Dicentrine positively affected the stability of G-quadruplex-forming oligonucleotides, whereas no relevant effect was observed in the control duplex, further corroborating the good G-quadruplex vs. duplex selectivity of this compound, as found using the G4-CPG assay.

Significant fluorescence quenching was observed upon the titration of Dicentrine with the investigated oligonucleotides, further confirming its strong binding to telomeric and oncogenic G-quadruplexes. Moreover, some preference for the hybrid telomeric G-quadruplex (K_b_ = (2.8 ± 1.2) × 10^6^ M^−1^) compared with the parallel oncogenic G-quadruplex (K_b_ = (0.8 ± 0.5) × 10^6^ M^−1^) was observed, in line with the affinity trend found using the G4-CPG assay. In addition, good G-quadruplex vs. duplex selectivity was found, considering the difference of one order of magnitude in their binding constants (K_b_ for the model duplex = (3.2 ± 0.4) × 10^5^ M^−1^)), in full agreement with the G4-CPG assay. Binding stoichiometries of 1:1, 1:2 and 1:5 were obtained for telomeric G-quadruplex/Dicentrine, oncogenic G-quadruplex/Dicentrine and control duplex/Dicentrine systems, respectively, proving that the binding to G-quadruplex structures was more specific compared with duplex DNA.

MD simulations allowed for a deeper insight into the interaction of Dicentrine with G-quadruplex structures. In the case of the telomeric G-quadruplex model, Dicentrine was proved to bind stably and persistently to the groove of the G-quadruplex structure in close proximity to G4-G6, T8 and G23-T25. On the other hand, in the case of the oncogenic G-quadruplex model, Dicentrine preferentially recognized the outer G-tetrad formed by G4, G8, G13 and G17 by displacing A3 and stacking on top of the G-tetrad in a sandwich-like conformation with G2. According to the 1:2 DNA/ligand binding stoichiometry of Dicentrine to the oncogenic G-quadruplex observed with fluorescence spectroscopy, the second Dicentrine molecule was found to bind to different sites of the G-quadruplex, including (i) the groove, including the loop T11-A12, by stacking on A12 and (ii) stacking on top of the first Dicentrine molecule on the G-tetrad at the 5′-end of the G-quadruplex in a sandwich-like conformation with G2.

Then, Dicentrine was evaluated for its anticancer activity and ability to target G-quadruplex structures in cells using biological assays. Altogether, the biological results highlighted the effectiveness of Dicentrine in promoting potent and selective antitumor activity by inducing cell cycle arrest through apoptosis. In agreement with the biophysical data, Dicentrine was found to promote the stabilization of G-quadruplex structures in cancer cells, mainly targeting and damaging telomeric DNA. 

Based on all these data, Dicentrine emerged as a strong and selective ligand of G-quadruplexes, even more effective and showing higher anticancer activity than its parent compound, Bulbocapnine. Furthermore, our results allow us to hypothesize that the previously proven anticancer activity of Dicentrine in leukemia cell line K562 in SCID mice could be associated with its ability to target cancer-related G-quadruplex structures in vivo. Overall, Dicentrine can be considered a promising and selective anticancer agent whose activity against cancer cells is well correlated with its ability to selectively target genomic and mainly telomeric G-quadruplexes, discriminating duplex DNA. It thus represents a valid lead compound for the design and evaluation of suitably improved analogs to be advanced to future in vivo studies aimed at the pharmacological development of a new class of potent and selective anticancer drugs that can overcome the toxicity problems associated with current therapeutics.

## Figures and Tables

**Figure 1 ijms-24-04070-f001:**
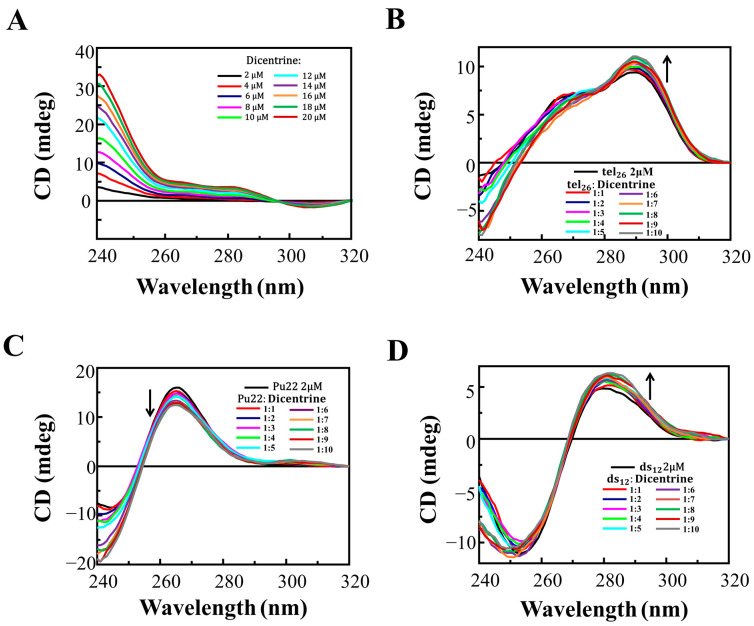
(**A**) CD spectra of solutions of Dicentrine (from 2 to 20 µM) in 5 mM KCl, 5 mM KH_2_PO_4_, 5% DMSO buffer (pH 7) and CD spectra of 2 μM solutions of (**B**) tel_26_ G-quadruplex, (**C**) Pu22 G-quadruplex and (**D**) ds_12_ duplex in 5 mM KCl, 5 mM KH_2_PO_4_, 5% DMSO buffer (pH 7) in the presence of increasing amounts of Dicentrine (up to 10 equivalents). Arrows indicate the CD changes on increasing the ligand concentration.

**Figure 2 ijms-24-04070-f002:**
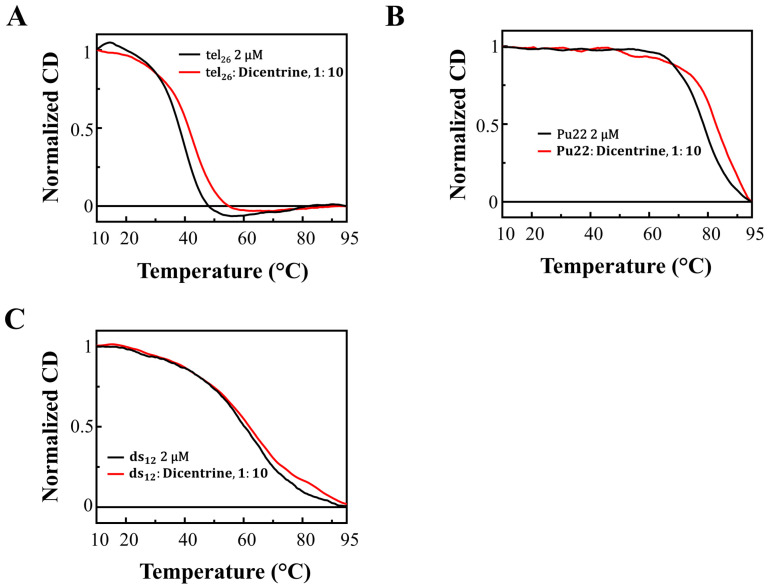
CD melting curves for (**A**) tel_26_ G-quadruplex, (**B**) Pu22 G-quadruplex and (**C**) ds_12_ duplex in the absence and presence of 10 molar equivalents of Dicentrine in 5 mM KCl, 5 mM KH_2_PO_4_, 5% DMSO buffer (pH 7), recorded at 290, 265 and 253 nm, respectively.

**Figure 3 ijms-24-04070-f003:**
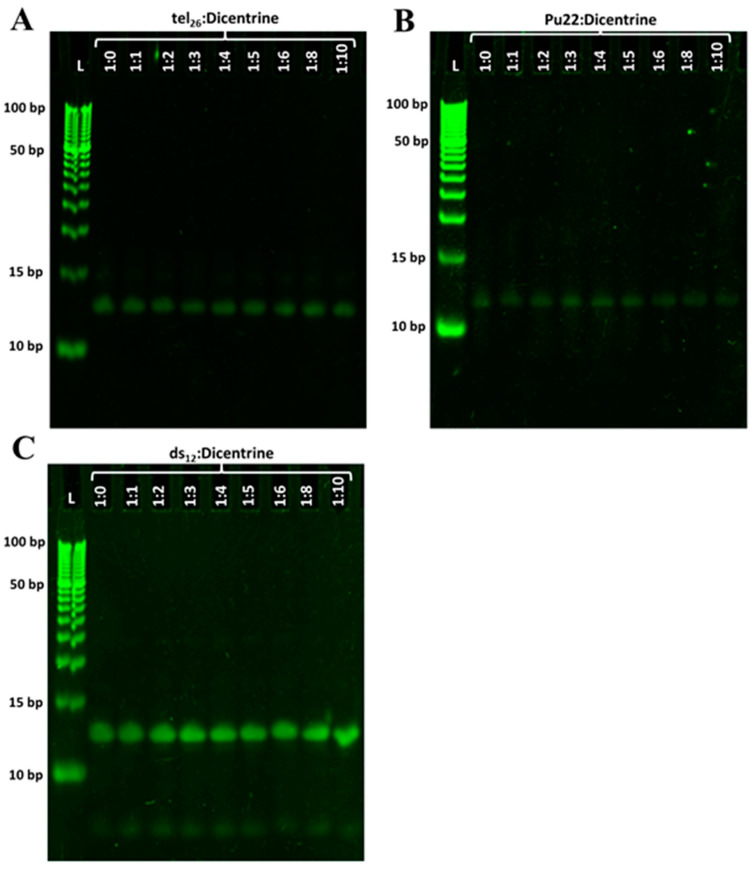
Native PAGE experiments: (**A**) tel_26_ G-quadruplex, (**B**) Pu22 G-quadruplex and (**C**) ds_12_ duplex samples were loaded at a 2 μM concentration in the absence and presence of different Dicentrine molar equivalents (from 1 to 10) and analyzed on 20% native PAGE. Lanes L: 5 bp DNA ladder. Gels were visualized with GelGreen™ staining.

**Figure 4 ijms-24-04070-f004:**
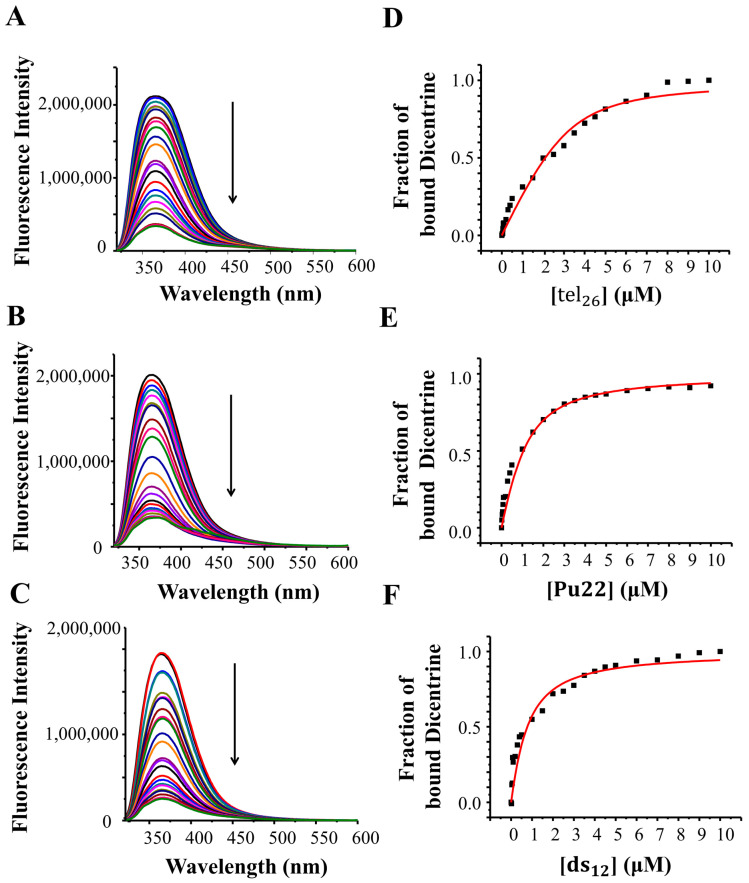
**Left**: Fluorescence emission spectra obtained by adding increasing amounts of (**A**) tel_26_ G-quadruplex, (**B**) Pu22 G-quadruplex and (**C**) ds_12_ duplex to 2 μM solutions of Dicentrine. Arrows indicate the fluorescence intensity changes upon increasing the DNA concentration. **Right**: Representative binding curves obtained by plotting the fraction of bound Dicentrine to (**D**) tel_26_ G-quadruplex, (**E**) Pu22 G-quadruplex and (**F**) ds_12_ duplex as a function of the DNA concentration. The black squares represent the data from a single experimental replicate; the red line represents the best fit obtained using an independent and equivalent-sites model.

**Figure 5 ijms-24-04070-f005:**
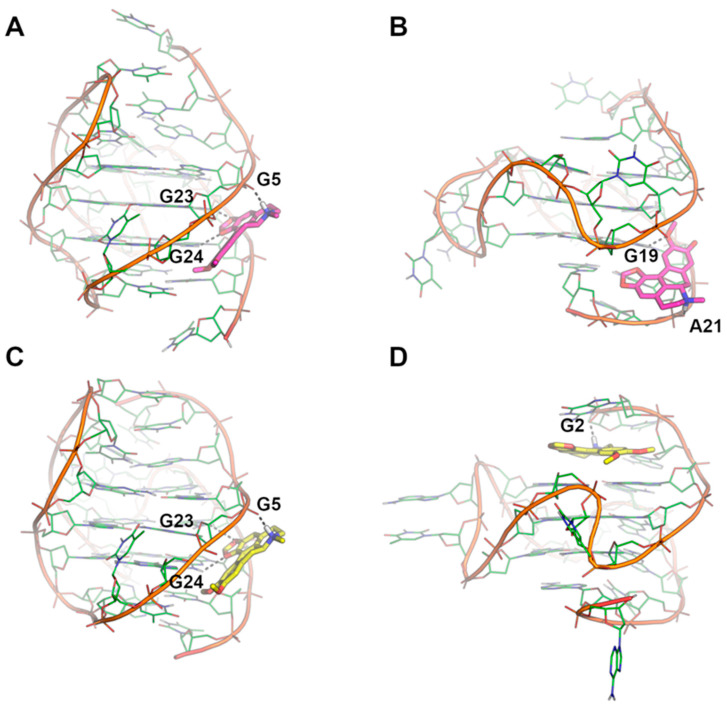
Binding modes of Dicentrine to tel_26_ and Pu22 G-quadruplexes as predicted by molecular docking and MD simulations. (**A**) Docking-based binding mode of Dicentrine to the first NMR model of tel_26_ (PDB-ID: 2JPZ) [38]. (**B**) Docking-based binding mode of Dicentrine to the first NMR model of Pu22 (PDB-ID: 1XAV) [39]. (**C**) Representative frame extracted from the MD trajectory of tel_26_/Dicentrine complex. (**D**) Representative frame extracted from the MD trajectory of Pu22/Dicentrine complex. G-quadruplexes are shown as green cartoon and lines. Dicentrine is shown using sticks and is colored magenta in the docking poses and yellow in the poses obtained by MD simulations. Polar contacts of Dicentrine are highlighted by black dashed lines; nucleotides involved in polar interactions with Dicentrine are labeled.

**Figure 6 ijms-24-04070-f006:**
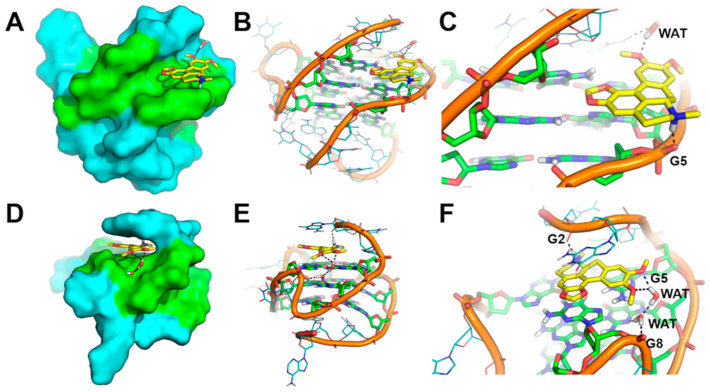
Representative MD structures describing the interaction between Dicentrine and tel_26_ and Pu22 G-quadruplexes. (**A**) Surface representation of tel_26_. (**B**) Cartoon representation of tel_26_. (**C**) Magnification of the binding mode of Dicentrine in the groove of tel_26_. (**D**) Surface representation of Pu22. (**E**) Cartoon representation of Pu22. (**F**) Magnification of the binding mode of Dicentrine on top of the G-tetrad of Pu22. Nucleotides from G-tetrads are colored green, other nucleotides are colored cyan. Dicentrine is shown as sticks and colored yellow. Water molecules bridging the ligand and G-quadruplexes are shown as sticks. Polar contacts are highlighted by black dashed lines. Nucleotides and water molecules (WAT) involved in polar interactions with Dicentrine are labeled in panels (**C**,**F**).

**Figure 7 ijms-24-04070-f007:**
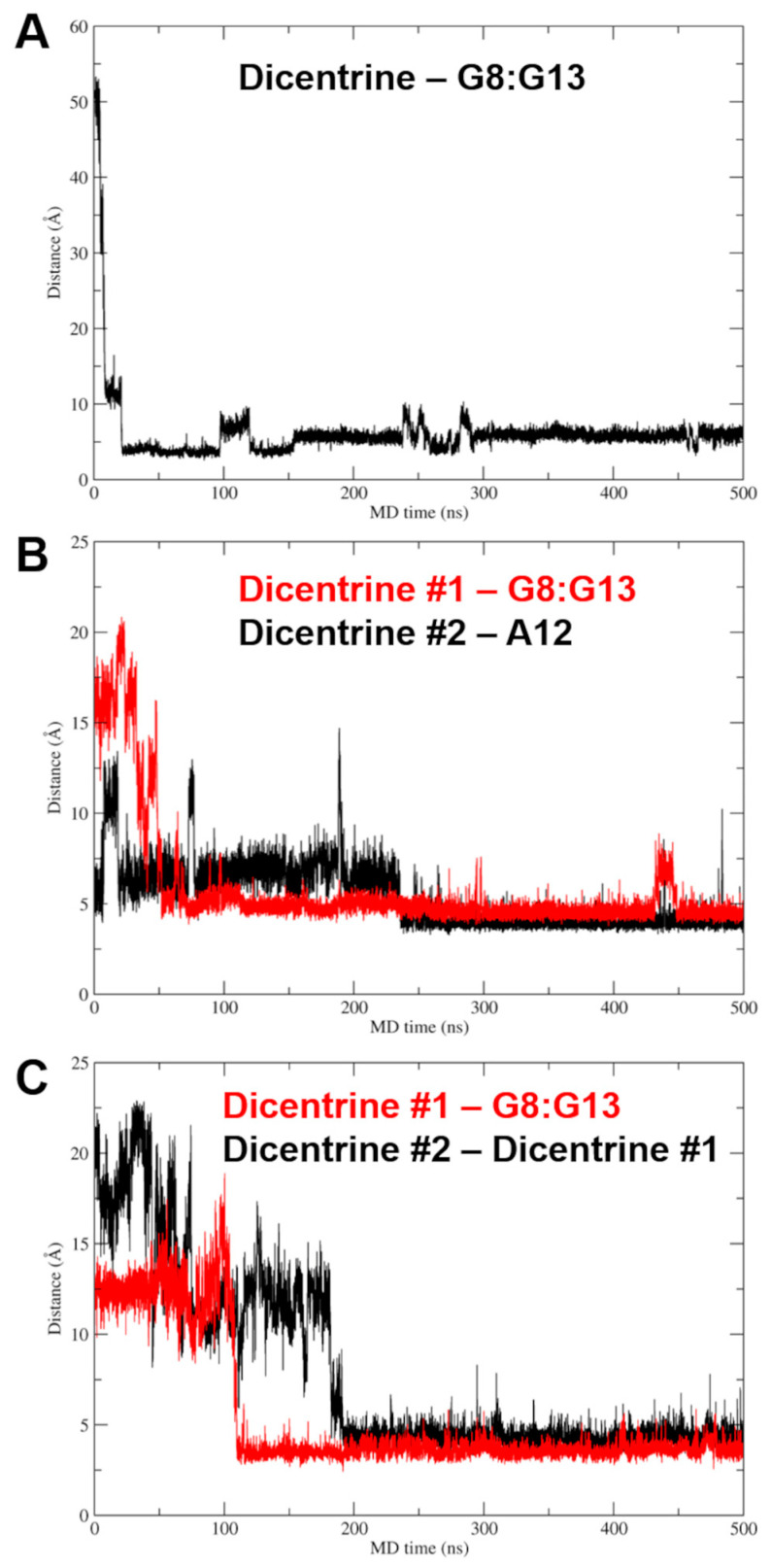
Plots of the distance between the mass center of Dicentrine and target nucleotides in MD simulations. (**A**) Distance between Dicentrine and the G8:G13 pair in the G-tetrad in the MD trajectory of the Pu22/Dicentrine complex with 1:1 stoichiometry. (**B**) Distance between Dicentrine #1 and the G8:G13 pair in the G-tetrad (red) and distance between Dicentrine #2 and the A12 mass center (black) in the first MD replica on the Pu22/Dicentrine complex with 1:2 stoichiometry. (**C**) Distance between Dicentrine #1 and the G8:G13 pair in the G-tetrad (red) and distance between Dicentrine #2 and the Dicentrine #1 mass center (black) in the second MD replica on the Pu22/Dicentrine complex with 1:2 stoichiometry.

**Figure 8 ijms-24-04070-f008:**
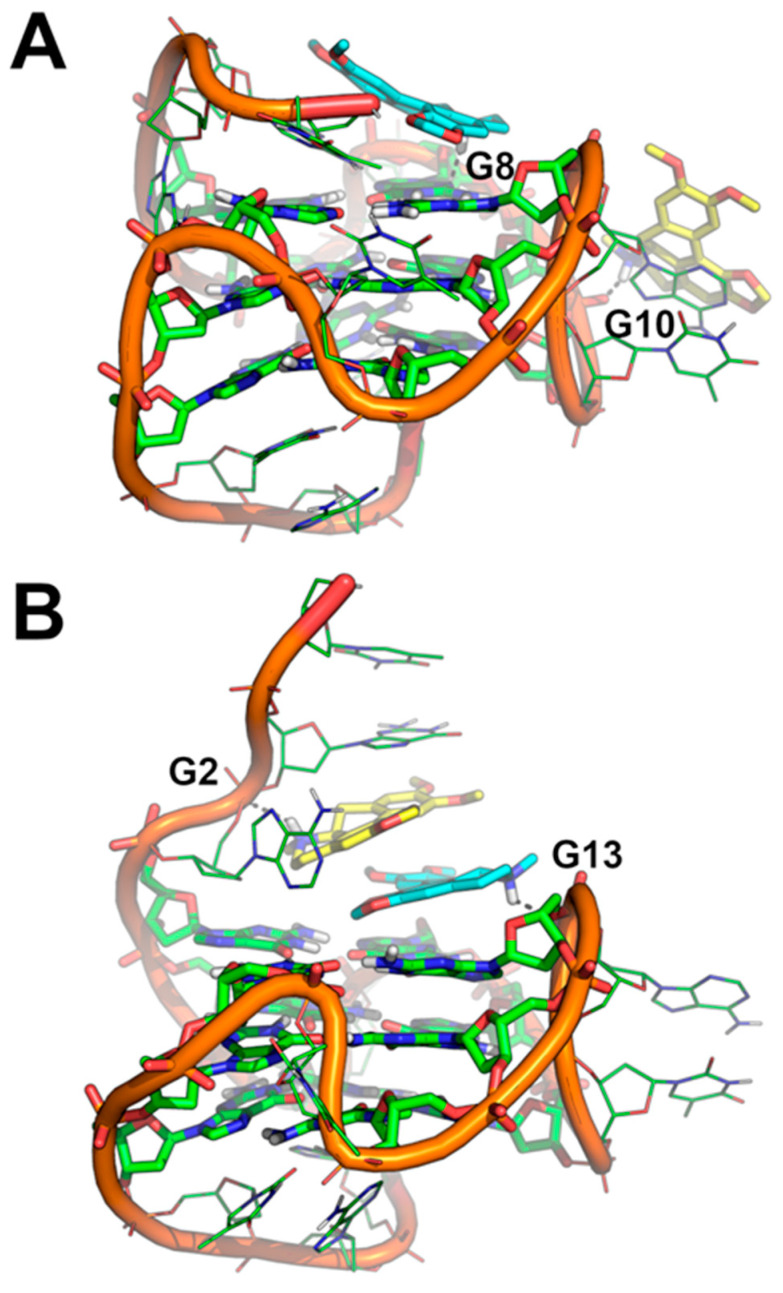
(**A**,**B**) Binding mode of two molecules of Dicentrine to the Pu22 G-quadruplex investigated using two independent replicas of MD simulations. The representative frame extracted by the MD trajectories through cluster analysis is shown. Pu22 is represented as green cartoon and lines; residues forming the G-tetrads are shown as sticks. Dicentrine #1 and Dicentrine #2 are shown as sticks and colored cyan and yellow, respectively. Polar contacts are highlighted by black dashed lines; nucleotides involved in polar interactions with Dicentrine are labeled.

**Figure 9 ijms-24-04070-f009:**
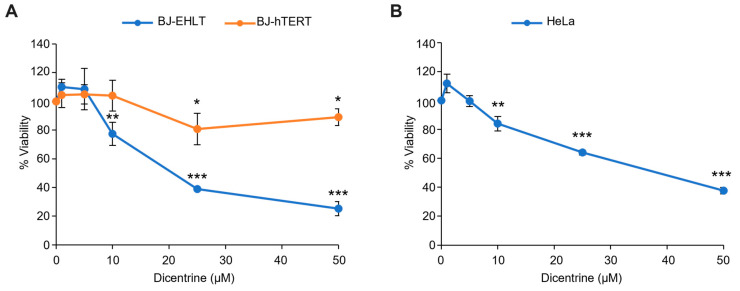
Effect of Dicentrine on cell viability. (**A**) BJ-EHLT (blue line) and BJ-hTERT (orange line) fibroblasts or (**B**) HeLa cells were treated for 72 h with increasing doses of Dicentrine (1, 5, 10, 25 and 50 µM) or with DMSO (negative control). Cell viability was determined with a colorimetric crystal violet assay. Results are expressed as the percentage of viable cells over the negative control. Histograms show the mean values ± SD of three independent experiments. * *p* < 0.05; ** *p* < 0.01; *** *p* < 0.001.

**Figure 10 ijms-24-04070-f010:**
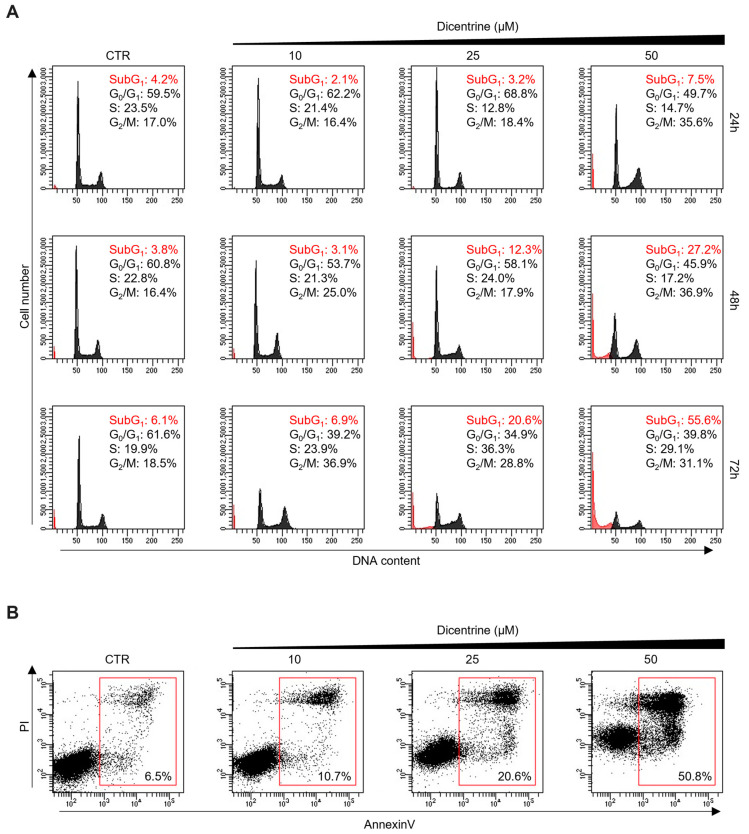
Dicentrine affects the cell cycle and induces apoptosis in cancer cells. HeLa cells were treated with DMSO (negative control) or increasing doses of Dicentrine (10, 25, 50 µM) for the indicated times and then processed for flow cytometry analysis. (**A**) Cells were stained with propidium iodide (PI), and an analysis of the cell cycle was performed. Quantification of the percentage of cells in subG_1_ and G_2_/M phases are reported in red and in black, respectively. (**B**) Two-dimensional scatter plots of Annexin V analysis performed on HeLa cells untreated or treated with growing doses of Dicentrine (10, 25, 50 µM) for 72 h. Red boxes evidence the populations of Annexin V-positive cells. One representative image is shown for each experiment.

**Figure 11 ijms-24-04070-f011:**
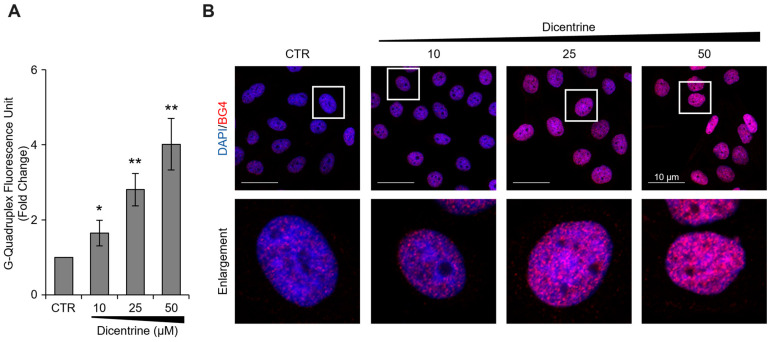
Biological evaluation of the G-quadruplex-stabilizing activity of Dicentrine. Immunofluorescence analysis of G-quadruplex structures in Hela cells treated for 24 h with DMSO (negative control) or with increasing doses of Dicentrine (10, 25, 50 µM). (**A**) Quantitative analysis of the anti-G-quadruplex signal. For each panel, at least 25 cells/conditions were analyzed using the ImageJ software, and the results were expressed as the fold change in the fluorescence intensity over the negative control. Histograms show the mean ± SD of three independent experiments performed in triplicate. * *p* < 0.05; ** *p* < 0.01. (**B**) ) *Upper Panels:* representative images of confocal sections (63×) used for the detection of G-quadruplex structures. For each experimental condition, merged images of DAPI counterstained nuclei (blue) and G-quadruplex structures (red) are shown. *Lower panels:* enlargements (4×) of a single cell from each of the upper panels (white squares) are shown. Scale bars (10 µm) are reported in the figures.

**Figure 12 ijms-24-04070-f012:**
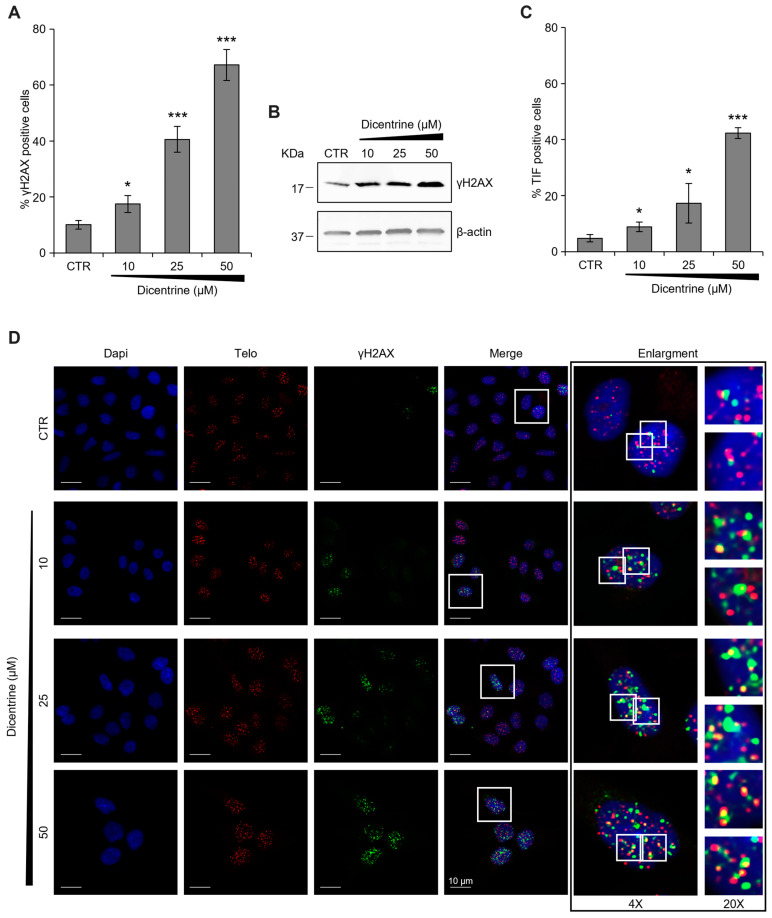
Dicentrine induces telomere-specific DNA damage. (**A**) HeLa cells were treated with DMSO (negative control) or with the indicated doses of Dicentrine (10, 25, 50 µM) for 24 h. Cells were processed for telomeric FISH combined with immunofluorescence, and the amount of DNA damage was evaluated. (**A**) Quantitative analysis of damaged cells. Results are expressed as a percentage of γH2AX positive cells over the negative control. (**B**) Western blot analysis of γH2AX levels. Β-actin protein levels were used as the loading control. (**C**) Quantitative analysis of the percentage of TIF-positive cells. Cells with at least four γH2AX-Telo colocalizations were scored as TIF-positive. (**D**) Representative images of confocal sections (63×) used for the quantitative analyses reported in (**A**,**C**). DAPI counterstained nuclei (blue), telomeres (red), γH2AX (green) and merged images are shown. For each experimental condition, 4× and 20× enlargements of merged fields are reported (white boxes). Scale bars (10 µm) are shown in the figures. Histograms show the mean ± SD of three independent experiments performed in triplicate. * *p* < 0.05; *** *p* < 0.001.

**Table 2 ijms-24-04070-t002:** Summary of the binding assay data obtained for the studied natural compounds using the G4-CPG assay. The bound ligand amounts are calculated as the difference between the initially loaded amount of ligand and the unbound ligand recovered by the washing solution and are expressed as a percentage of the quantity initially loaded on each support. The errors associated with the reported percentages are within ±2%.

Compound	Bound Ligand (%)			
	Nude CPG	CPG-tel_26_	CPG-c-myc	CPG-ds_27_
Canadine	21	29	20	4
D-Glaucine	0	2	1	0
Dicentrine	0	67	47	30
Deguelin	0	0	0	0
Millettone	8	8	8	8

**Table 3 ijms-24-04070-t003:** Melting temperature (T_m_) values of tel_26_ G-quadruplex, Pu22 G-quadruplex and ds_12_ duplex in the absence or presence of Dicentrine (10 molar equivalents) as measured by CD melting experiments. ΔT_m_ = T_m_(DNA/ligand, 1:10)−T_m_(free DNA).

	T_m_ ± 1 (°C)	ΔT_m_ (°C)
tel_26_	40	-
tel_26_/Dicentrine, 1:10	43	+3
Pu22	79	-
Pu22/Dicentrine, 1:10	84	+5
ds_12_	63	-
ds_12_/Dicentrine, 1:10	64	+1

**Table 4 ijms-24-04070-t004:** Delta energy of the binding of Dicentrine to the Pu22 and tel_26_ G-quadruplexes as predicted using the MM-GBSA approach.

Compound	Binding Energy to 5′-End G-Tetrad of Pu22 (kcal/mol ± SEM)	Binding Energy to A12 (kcal/mol ± SEM)	Binding Energy to Dicentrine #1 (kcal/mol ± SEM)	Binding Energy to tel_26_ (kcal/mol ± SEM)
Dicentrine #1	−27.43 ± 4.03 (R1) ^a^ −26.85 ± 4.65 (R2) ^a^	-	-	−26.34 ± 3.68
Dicentrine #2	-	−16.13 ± 1.77	−19.65 ± 2.75	-

^a^ R1 = MD replica #1; R2 = MD replica #2.

## Data Availability

Not applicable.

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
