# Peer review of "Selective Targeting of Cancer-Related G-Quadruplex Structures by the Natural Compound Dicentrine"

_ijms, 2023, doi:10.3390/ijms24044070_

Round 1

Reviewer 1 Report

The manuscript entitled "Selective targeting of cancer-related G-quadruplex structures by the natural compound Dicentrine" provides an interesting perspective on the development and usage of Dicentrine as a potential anticancer drug by targeting and stablizing the G-qudraplexes. The authors have used a wide array of biochemical and biophysical assays to prove the stabilisation of G-qudraplexes upon binding of Dicentrine. In cell studies shows the cell cycle arrest and induced apoptosis of cancerous cells upon treatment of Dicentrine, while IF studies further provide evidence about the effect of Dicentrine in inducing DNA damage in Hela Cells at the telomeres. Overall, the authors have provided a comprehensive studies about the advantages of Dicentrine , a natural compound (which in itself is a major strength of this study) as a future anticancer therapeutics. The manuscript should be accepted for publication however, I would like the authors to address some of my minor queries/concern before that.

1) I would advise the authors to proofread the whole MS again. Minor English editing is required. for e.g. somewhere it was written "greater amount of Dicnetrine" where the word "greater"
 should be replaced by "increased"

2)Figure 4: How many replicates for this particular experiments were performed. This should be clearly mentioned as I didn't see any standard error bars in the binding analysis study

3) In which Groove of the G-quadraplex the docking was carried out. Also what were the binding energies? 

4) Is it possible to represent Dicentrine as stick model with intermolecular interactions with duplexes? Also all the figures (5,6,8) needs to be properly labelled in terms of nucleotides interacting with Dicentrine for better understanding.

5) Label the interacting nucleotides in the figures too. Also, I am unable to locate any water mediated H bonding.

6) Line 324: Again highlight  all the nts in the figure. It will make it easier to understand. Also what are the binding energies at each step during the simulation of dicentrine to Pu22 and how they are comparable to tel26?

7) Line 351: I think there is a need to specify the groove of dicentrine binding, otherwise it is really getting confusing. Also, it would be good to provide a simple cartoon figure depicting the binding site of dicentrine in both tel26 and Pu22 to avoid this confusion. This could be provided as a supplimentary figure.

 8) Figure 8: Is it possible that both the dicentrine molecule can bind simultaneously to both sides? Also did the authors found any second binding site in tel26?

9) Line 378-386: These energies should be provided in table. 

10) Line 424: Just curious whether BJ-EHLT can be used instead of HeLa cells in FACS or not? If yes, then is there any reason why it was not used for FACS study?

11) Can authors provide a small comment on advantages of  Dicentrina as a potential anticancer drug over the current therapies. This should be included in discussion sections and will definitely increase the significance of this study.

Reviewer 2 Report

The authors of this study tested some naturally occurring compounds to target G-quadruplexes occurring in cancer-related nucleic acids.

As a result of screening, one of the molecules was investigated further. Decentrine was shown to be effective in G-quadruplex binding. What is correlated with the anticancer activity of the compound.

The article is engaging and interesting, but I would like the authors to highlight more the results that support the statement from the title that decentrine is selective.

Maybe I've missed something, but the authors used a keyword: ribozymes. Can I ask the authors to justify this choice?
